# UltraFeedback: Boosting Language Models with High-quality Feedback

## Abstract

Reinforcement learning from human feedback (RLHF) has become a pivot technique in aligning large language models (LLMs) with human preferences. In RLHF practice, preference data plays a crucial role in bridging human proclivity and LLMs. However, the scarcity of diverse, naturalistic datasets of human preferences on LLM outputs at scale poses a great challenge to RLHF as well as feedback learning research within the open-source community. Current preference datasets, either proprietary or limited in size and prompt variety, result in limited RLHF adoption in open-source models and hinder further exploration. In this study, we propose UltraFeedback, a large-scale, high-quality, and diversified preference dataset designed to overcome these limitations and foster RLHF development. To create UltraFeedback, we compile a diverse array of instructions and models from multiple sources to produce comparative data. We meticulously devise annotation instructions and employ GPT-4 to offer detailed feedback in both numerical and textual forms. UltraFeedback establishes a reproducible and expandable preference data construction pipeline, serving as a solid foundation for future RLHF and feedback learning research. Utilizing UltraFeedback, we train various models to demonstrate its effectiveness, including the reward model UltraRM, chat language model UltraLM-13B-PPO, and critique model UltraCM. Experimental results indicate that our models outperform existing open-source models, achieving top performance across multiple benchmarks.

## 1 Introduction

Large language models (LLMs), represented by ChatGPT (OpenAI, 2022) and GPT-4 (OpenAI, 2023), have demonstrated proficiency in generating fluent text as well as solving various language-oriented tasks. Trained on massive corpora through likelihood maximization techniques, these LLMs have exhibited remarkable generalization and equipped the ability to execute diverse tasks in response to user directives (Ouyang et al., 2022; Wei et al., 2022; Sanh et al., 2022). Unfortunately, relying solely on likelihood maximization during training leads to well-known issues - LLMs may generate convincing but incorrect or unsafe content that deviates from human preferences (Stiennon et al., 2020; Ouyang et al., 2022; Perez et al., 2022). To further align LLMs with human preferences, *reinforcement learning from human feedback* (RLHF) (Ouyang et al., 2022; Askell et al., 2021; Bai et al., 2022a; Touvron et al., 2023b) has been introduced and widely adopted by leading corporations.

RLHF builds upon preference data, which rates and compares different responses given the same prompt. Typically, RLHF trains a reward model on preference data and then applies RL algorithms such as Proximal Policy Optimization (PPO) (Schulman et al., 2017) on LLMs to optimize the rewards (OpenAI, 2022; 2023; Touvron et al., 2023b; Bai et al., 2022a). While proprietary models have largely capitalized on RLHF's potential to produce outputs that are both more useful and safer, a significant gap persists in the open-source community. Such a gap is mainly characterized by the lack of high-quality, publicly available preference datasets.Existing preference datasets are either small in scale (Wu et al., 2023), limited on specific tasks (Stiennon et al., 2020; Nakano et al., 2021), or in short of meticulous preference annotations (Askell et al., 2021). As a result, few open-source models adopt RLHF as it demonstrates marginal gains, which critically hinders successful RLHF practice and further research.

To facilitate the data shortage issue and unleash the power of RLHF on open-source models, we construct ULTRAFEEDBACK, a large-scale, high-quality, and diversified preference dataset. Compared with existing preference datasets, ULTRAFEEDBACK presents the following advantages: (1) Scale. ULTRAFEEDBACK is **the largest non-community-labeled open-source preference dataset** with 64k instructions and 4 responses per instruction, leading to over 340k comparison data. (2) Diversity. To ensure both instructions and responses in ULTRAFEEDBACK are diverse, we extensively curate the instruction pool and model pool from different sources. For further response diversity, we design over forty principle prompts to induce various model behaviors. (3) Fine-grained annotations. We carefully curate a fine-grained annotation instruction, which contains 4 different aspects, namely **instruction-following, truthfulness, honesty, and helpfulness** to comprehensively assess the responses. We then ask GPT-4 to annotate the collected samples based on the instructions, which have been proven to produce high-quality annotations (Bai et al., 2022b; Lee et al., 2023). Besides numerical preference scores, we also ask GPT-4 to provide rigorous explanations for the annotation and suggestions for each response. The textual feedback enables the training of a critique model, which assists in automatic evaluation and model improvement.

We comprehensively validate the advantage of ULTRAFEEDBACK and further boost open-source models with the dataset. By fine-tuning a LLaMA2-13B model (Touvron et al., 2023b), we build a state-of-the-art reward model UltraRM, which significantly outperforms existing open-source reward models. Based on UltraRM, we enhance a powerful open-source model UltraLM (Ding et al., 2023; Touvron et al., 2023a) with best-of-$n$ sampling and PPO. Experiments show that both strategies boost the model dramatically. Moreover, since ULTRAFEEDBACK provides detailed textual feedback, we also fine-tune a model that could critique model responses automatically. Our critique model, UltraCM, generates reasonable and detailed comments on various tasks.

## 2 RELATED WORK

**RLHF for LLMs.** In the past few years, reinforcement learning for language models has been extensively studied by leading corporations such as OpenAI, Anthropic, and Google DeepMind. Early works applied RLHF to enhance language models on specific tasks such as summarization (Stiennon et al., 2020) and question answering (Nakano et al., 2021; Menick et al., 2022). Afterward, RLHF has been widely adopted on advanced general language models (Ouyang et al., 2022; Bai et al., 2022a; Glaese et al., 2022; OpenAI, 2022; 2023; Touvron et al., 2023b) to align with human values, and the RLHF versions are mostly more helpful, more truthful, and less harmful (Perez et al., 2022; Ouyang et al., 2022). However, these top companies seldom open-source their preference datasets and reward models, hindering other researchers from reproducing their results. The open-source research community also put efforts into demystifying RLHF, which dive into the RL implementation details (Zheng et al., 2023b; Song et al., 2023b), design novel algorithms (Gulcehre et al., 2023; Rafailov et al., 2023; Song et al., 2023a) to utilize preference data more efficiently, or align LLMs with various types of feedbacks (Yao et al., 2023; Wang et al., 2023c; Liu et al., 2023). To compare, our work explores **how to automatically construct high-quality feedback data**, which is fundamental yet usually overlooked in the literature.

**Datasets for LLM Alignment.** The importance of data scalability and quality has been widely recognized in the literature on instruction tuning (also known as SFT). Early works collected various NLP tasks or real user conversations to conduct instruction tuning and observed that LLMs could generalize well across different tasks (Wei et al., 2022; Sanh et al., 2022; Wang et al., 2022; Ouyang et al., 2022). After the release of ChatGPT, most recent research on SFT emphasized the importance of data construction and reached conclusions that scalability, diversity, as well as quality, are vital for the final performance (Ding et al., 2023; Taori et al., 2023; Chiang et al., 2023; Xu et al., 2023). However, when it goes to the RLHF stage, the importance of data engineering has not been well illustrated. Among current preference datasets, some of them focus on specific tasks (e.g. summarization (Stiennon et al., 2020), search-based question answering (Nakano et al., 2021), safety-oriented scenarios (Ji et al., 2023), and math problems (Lightman et al., 2023)), thus cannot boost general chat models. Some datasets are small in scale (Wu et al., 2023; Wang et al., 2023c) or provide only community votes as coarse-grained preferences (Ethayarajh et al., 2022; Askell et al., 2021). Therefore, there lacks a large general-purpose preference dataset with diverse instructions and fine-grained annotations in the open-source community, which motivates us to construct ULTRAFEEDBACK.

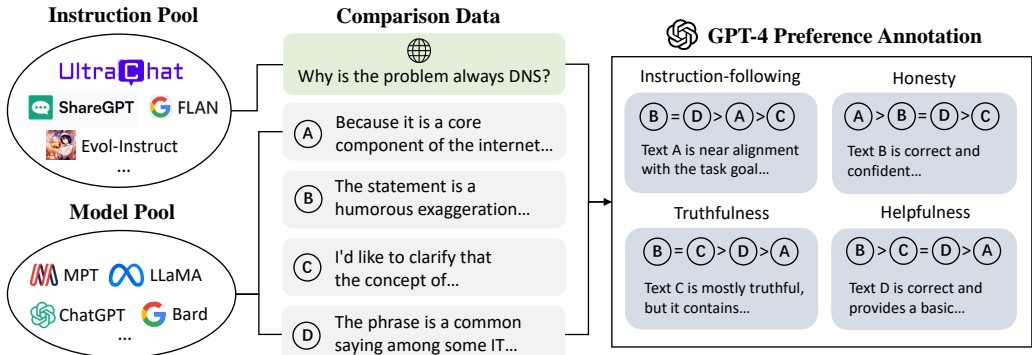

Figure 1: ULTRAFEEDBACK construction process. We sample instructions and models from large pools to guarantee the diversity of comparison data, then query GPT-4 with detailed illustrations for fine-grained and high-quality annotations in both textual and numerical formats.

## 3 ULTRAFEEDBACK

To alleviate the shortage of preference data, we construct ULTRAFEEDBACK and illustrate the process in Figure 1. In this section, we will describe how we build ULTRAFEEDBACK (Section 3.1) in detail and introduce models powered by it (Section 3.4).

### 3.1 CONSTRUCTION PROCESS

We start by identifying four essential requirements of LLMs in practical use: (1) **Follow Instructions**: LLMs should comprehend the intention of human instructions and respond to humans without deviating from the requirements, including both task goals and extra restrictions. (2) **Helpful and Informative**: LLMs should provide useful and correct answers to successfully address the given problems. (3) **Truthful**: LLMs' output should be grounded in the instructions and real-world knowledge, not fabricating any facts without references or introducing any self-contradiction. (4) **Honesty**: LLMs should know what they (don't) know and express uncertainty when they are in waver towards the given problem.

To align LLMs with all the characteristics, the reward model should be trained on preference data of four aspects. Therefore, we are motivated to construct a diversified preference dataset in which both instructions and model completions contain contents of different aspects. To this end, we sample instructions that are challenging to certain aspects of LLMs' capability and subsequently prompt various models to behave correctly in the corresponding aspect.

**Instruction Sampling.** To yield varying levels of completion, we investigate datasets that assess certain abilities of LLMs. Specifically, we include all instructions from TruthfulQA (Lin et al., 2022) and FalseQA (Hu et al., 2023), randomly sample 10k instructions from Evol-Instruct (Xu et al., 2023) and UltraChat (Ding et al., 2023) respectively, and sample 20k from ShareGPT (Chiang et al., 2023). For FLAN (Longpre et al., 2023), we adopt a stratified sampling strategy (Mukherjee et al., 2023), randomly picking 3k instructions from the "CoT" subset and sampling 10 instructions per task for the other three subsets, while excluding those with overly long instructions. We then conduct a data contamination detection and filter out those that overlap with AlpacaEval, UltraChat test set, and Evol Instruct test set (Appendix B). Finally, we sample 63, 967 instructions from the six publicly available high-quality datasets.

**Model Sampling.** To guarantee the collected responses are also dissimilar and well-distributed, we select different models at all levels to complete the instructions, with varying model sizes, architectures, and training data. We set up a pool of 17 models: (1) For commercial models, we choose GPT-4, gpt-3.5-turbo (ChatGPT), and Bard [1]; (2) For LLaMA-series, we choose UltraLM-13B/65B (Ding et al., 2023), WizardLM-7B-v1.1/13B-v1.2/70B-v1.1 (Xu et al., 2023), Vicuna-33B-v1.3 (Chiang et al., 2023), LLaMA2-7B/13B/70B-Chat (Touvron et al., 2023b), and

---

[1] https://bard.google.com/

Table 1: Statistics of existing preference and critique datasets. The average length refers to the number of tokens.

| Dataset | # Convs | Avg. Len. of Instruction | Avg. Len. of Completion | Avg. Len. of Critique | Fine-Grained? | Feedback Format | # Comparisons | # Critique |
|---|---|---|---|---|---|---|---|---|
| *Preference Dataset* | | | | | | | | |
| OASST1 | 35,905 | 167.6 | 221.1 | - | ✗ | Scalar | 17,966 | - |
| OpenAI WebGPT | 38,925 | 50.9 | 188.2 | - | ✗ | Scalar | 19,578 | - |
| Anthropic Helpful | 118,263 | 185.7 | 94.6 | - | ✗ | Ranking | 118,263 | - |
| OpenAI Summ. | 60,674 | **326.4** | 36.6 | - | ✓ | Scalar | 92,858 | - |
| QA Feedback | 11,378 | 155.8 | 107.9 | - | ✓ | Scalar | 17,118 | - |
| *Critique Dataset* | | | | | | | | |
| SelFee | 178,331 | 100.3 | 243.9 | 89.4 | ✓ | Text | - | **316,026** |
| Shepherd | 1,316 | 95.3 | 97.6 | 67.2 | ✓ | Text | - | 1,317 |
| ULTRAFEEDBACK | **255,864** | 185.1 | **305.3** | 143.1 | ✓ | Scalar & Text | **340,025** | 255,864 |

Alpaca-7B (Taori et al., 2023); (3) For Non-LLaMA series, we choose MPT-30B-Chat (MosaicML, 2023), Falcon-40B-Instruct (Almazrouei et al., 2023), StarChat (Tunstall et al., 2023), and Pythia-12B (Biderman et al., 2023). We randomly sample four different models from the pool to complete each instruction.

**Principle Sampling.** Following Sun et al. (2023) and Mukherjee et al. (2023), we define a set of principles for each aspect to prompt models to consider expected aspects in their responses. We hand-craft one principle for each aspect and then invoke GPT-4 to curate another ten based on the human-written example. According to dataset characteristics, we manually set principle prompts for each data source. For each completion, we randomly sample a corresponding principle and add it to the system prompt to induce model behaviors. The principles can be found in Appendix F.1.

## 3.2 ANNOTATION DETAILS

After generating $255,864$ model completions based on the $63,967$ instructions, we employ GPT-4 to provide two types of feedback for each completion: (1) scalar scores that indicate the fine-grained quality regarding multiple aspects, and (2) textual critique that gives detailed guidance on how to improve the completion.

**Preference Annotation.** To avoid subjective annotation, we provide detailed documentation of every aspect for GPT-4 to reference. In each document, we briefly introduce the definition of each aspect and then elaborate on expected behaviors from scores 1 to 5. (See Appendix F.2 for an example). To reduce randomness in GPT-4 annotation, we wrap one instruction and all its completions into the prompt and ask GPT-4 to score four completions simultaneously, avoiding GPT-4 applying inconsistent criteria to rate completions across different runs.

**Critique Generation.** Besides scalar reward, we also seek textual critique from GPT-4. We prompt GPT-4 to act as a tutor and provide detailed suggestions specified for each completion to help models improve rather than propose answers directly. Different from the above comparison-oriented annotations, critique prompts are generated separately from an overall perspective for each completion. We also prompt GPT-4 to give an overall score of each completion at the end of the critique to assess completion quality. The prompts can be found in Appendix F.2.

## 3.3 DATASET STATISTICS

We compare ULTRAFEEDBACK with current open-source datasets in Table 1. ULTRAFEEDBACK stands out to be the largest one among all preference and critique datasets, which is at least twice as large as other datasets. Also, its completions and critiques are the longest. Moreover, we highlight that ULTRAFEEDBACK is **the only dataset that provides both scalar preferences and textual feedback**, enabling it to serve as a preference and critique dataset simultaneously. Overall, ULTRAFEEDBACK outperforms previous datasets in both scale and diversity, and we also validate its high quality by experiment in Section 4.

### 3.4 ULTRAFEEDBACK-POWERED MODELS

Based on ULTRAFEEDBACK, we develop UltraRM, an advanced open-source reward model that provides preferences for AI responses given user instructions. Additionally, we train a critique model UltraCM from the textual feedback in ULTRAFEEDBACK. UltraCM could interact with human and AI assistants more flexibly in text.

**UltraRM.** For reward modeling, we train UltraRM based on LLaMA2-13B (Touvron et al., 2023b). Specifically, we train three versions of UltraRM. Firstly, to validate the quality of ULTRAFEEDBACK, we train UltraRM-UF with *merely* the fine-grained version of this dataset, i.e. averaging the preference scores in each aspect to get a final reward score. Then, to build a stronger reward model to boost RLHF, we mix several open-source datasets with ULTRAFEEDBACK to train UltraRM. The open-source datasets include Stanford SHP [2], OpenAI Summarization (Stiennon et al., 2020), and Anthropic Helpful (Bai et al., 2022a). To examine the effectiveness of the fine-grained scores compared with overall scores, we train UltraRM-Overall with the assessment ratings in critique generation. Note that the training data for UltraRM-Overall also mixed open-source data as same as UltraRM. The details for dataset processing can be found in Appendix D.1.

We keep the training strategy, including loss objective and training hyperparameters, exactly the same as Touvron et al. (2023b). We construct each comparison pair as a binary selection, with one completion being chosen and the other rejected. We optimize the reward model to select preferred completion by minimizing the binary ranking loss:

$$\mathcal{L}_{\text{ranking}} = -\log\left(\sigma\left(r_\theta\left(x, y_c\right) - r_\theta\left(x, y_r\right) - m(r)\right)\right) \quad (1)$$

where $\theta$ represents the reward model, $r_\theta\left(x, y_c\right)$ is its scalar reward prediction towards the chosen text, $r_\theta\left(x, y_r\right)$ is that towards the rejected text, and $m\left(r\right)$ is the absolute difference between the annotated scalar score of two texts. We set the $m\left(r\right) = 0$ for datasets with only preference rankings and normalize the margins to $(0, 1]$ to avoid training instability due to a mismatch between the score scales of the two datasets.

Following Touvron et al. (2023b), we train the 13B reward model for one epoch with the batch size being 512 pairs (i.e., 1024 completions) and the learning rate being 1e-5. We adopt the cosine learning rate decay strategy with a warm-up ratio of 3% and a final learning rate of 1e-6.

**UltraCM.** We also train a critique model stemming from UltraFeedback to boost future research in learning from feedback (Wang et al., 2023d). UltraCM has the same initialization as UltraRM but is trained solely on UltraFeedback critique data, i.e. $255, 864$ textual feedback in total. Given a response, we fine-tune the model to give a corresponding critique that judges the response, figures out flaws, and provides suggestions for improvement. We train LLaMA2-13B for two epochs with a batch size of 256 and a learning rate of 2e-5. We adopt the same learning rate scheduler as reward modeling.

## 4 EXPERIMENTS

To further validate the quality of ULTRAFEEDBACK, we first evaluate UltraRM on preference benchmarks in Section 4.1. Next, we test UltraRM in enhancing existing chat language models with two strategies, namely best-of-$n$ sampling (Section 4.2) and reinforcement learning (Section 4.3). Finally, we evaluate the feedback quality of UltraCM in Section 4.4.

### 4.1 REWARD MODELING

**Setup.** To evaluate how UltraRM aligns with human preference, we conduct experiments on four preference datasets, OpenAI WebGPT (Nakano et al., 2021), OpenAI Summarization (Stiennon et al., 2020), Anthropic HH-RLHF (Bai et al., 2022a), and Stanford SHP. On each dataset, we calculate the rewards of two responses for one prompt and predict which one is more preferred. We compare our UltraRM-UF, UltraRM-Overall, and UltraRM with open-source baselines, including Moss (Zheng et al., 2023b), Ziya (IDEA-CCNL, 2021), OASST [3], and SteamSHP [4]. We also report the results in LLaMA2 (Touvron et al., 2023b), although their reward models are not released.

---

[2] https://huggingface.co/datasets/stanfordnlp/SHP
[3] https://huggingface.co/OpenAssistant/reward-model-deberta-v3-large-v2
[4] https://huggingface.co/stanfordnlp/SteamSHP-flan-t5-xl

Table 2: Reward modeling accuracy (%) results. We compare our UltraRM with baseline open-source reward models. LLaMA2 results are taken from Touvron et al. (2023b). The highest results are in **bold** and the second highest scores are underlined.

| Model | Backbone Model | Open? | Anthropic Helpful | OpenAI WebGPT | OpenAI Summ. | Stanford SHP | Avg. |
|---|---|---|---|---|---|---|---|
| **Moss** | LLaMA-7B | ✓ | 61.3 | 58.1 | 59.0 | 54.6 | 58.1 |
| **Ziya** | LLaMA-7B | ✓ | 61.4 | 61.8 | 60.3 | 57.0 | 60.5 |
| **OASST** | DeBERTa-v3-large | ✓ | 67.6 | - | 71.8 | 53.9 | - |
| **SteamSHP** | FLAN-T5-XL | ✓ | 55.4 | 62.6 | 48.4 | 51.6 | 58.1 |
| **LLaMA2 Helpfulness** | LLaMA2-70B | ✗ | **72.0** | - | **75.5** | **80.0** | - |
| **UltraRM-UF** | LLaMA2-13B | ✓ | 66.7 | 65.1 | 66.8 | 68.4 | 66.8 |
| **UltraRM-Overall** | LLaMA2-13B | ✓ | 71.0 | 62.0 | 73.0 | 73.6 | 69.9 |
| **UltraRM** | LLaMA2-13B | ✓ | 71.0 | **65.2** | 74.0 | 73.7 | **71.0** |

**Results.** The preference prediction accuracy results are reported in Table 2. As we can see, the UltraRM series outperform baseline reward models except for the closed LLaMA2 reward model (much larger) by a large margin, indicating that UltraRM series are the **best open-source reward models**. Notably, our UltraRM-UF which does not train on any open-source preference datasets, also surpasses all other baselines. This result validates the high quality of ULTRAFEEDBACK, which enables strong out-of-distribution generalization. On average, UltraRM-UF outperforms open-source baseline models by over 6.3 percent in accuracy, while UltraRM-Oervall and UltraRM achieve 3.1 and 4.2 percent more improvement respectively.

We highlight that the OpenAI WebGPT dataset has no training and test splits, and neither most baselines nor we train reward models on this dataset[5], making it a fair benchmark to evaluate the generalization ability of reward models. Obviously, UltraRM series are significantly better, reaching 2.6% absolute points improvement over baselines. Another intriguing finding is that adding open-source datasets has a minor effect on the WebGPT dataset, which again proves the transferability advantage of ULTRAFEEDBACK. On another benchmark Stanford SHP, UltraRM also achieves remarkable performance.

A noteworthy finding is that, despite exhibiting comparably on the other three datasets, UltraRM-Overall discernably lags behind UltraRM-UF and UltraRM on WebGPT. There can be two potential explanations for this observation. First, fine-grained annotation, which scores model outputs from different aspects respectively, provides a more precise assessment for each completion than aggregating evaluation into an overall number. Second, in the overall quality annotation process, each sample is sent to GPT-4 separately whereas, in fine-grained rating, all four completions are scored at the same time, which may provide GPT-4 with cross-references and prevent it from applying inconsistent standards, reducing the impact of randomness. These superiorities demonstrate the high quality of our fine-grained preference data, and we advocate future work to adopt the fine-grained annotation schema and rate multiple completions at one time.

## 4.2 BEST-OF-$n$ EXPERIMENTS

**Setup.** To verify that our UltraRM could serve as a good indicator of response quality, we conduct best-of-$n$ experiments. On the AlpacaEval benchmark, we randomly sample 16 examples from the original UltraLM-13B and calculate their corresponding rewards. Thereafter, we select the best-of-$\{1, 2, 4, 8, 16\}$ responses and calculate their scores. The sampling parameters are set to temperature $= 1$ and top-$p = 1$.

**Results.** We present results in Figure 2. Apparently, we can see the win rate on AlpacaEval increases proportionally with rewards. This validates that our UltraRM gives rigorous rewards that reflect the overall response quality. Notably, the best-of-$n$ sampling strategy is surprisingly effective. The initial UltraLM-13B model achieves a 76.53% win rate for a single sampling, and a simple best-of-2 sample increases the win rate to 84.64%. With more samples, we can get even more high-quality responses,

---

[5]The OASST and LLaMA2 Helpfulness reward model used WebGPT dataset for training. To prevent data leakage, we do not report their performance on WebGPT.

and the final best-of-16 win rate hits $91.54\%$. The best-of-$n$ sampling is universally applicable across models and tasks, which enhances models without training. Please refer to Appendix E.2 for cases.

### 4.3 PPO EXPERIMENTS

**Setup.** Given the state-of-the-art UltraRM, we aim to push the upper bound of open-source chat language models with RLHF. Specifically, we perform PPO over UltraLM-13B (Ding et al., 2023) to get its PPO version, UltraLM-13B-PPO. We tune UltraLM for 80 iterations on the ULTRAFEEDBACK prompts. In each iteration, we collect 512 samples and update the policy model with a mini-batch size of 64. The learning rate is fixed at 1e-6.

**Baselines.** We compare UltraLM-13B-PPO with leading open-source models and proprietary models, including LLaMA2-Chat (Touvron et al., 2023b), Vicuna (Chiang et al., 2023), WizardLM (Xu et al., 2023), OpenChat (Wang et al., 2023a), and ChatGPT (OpenAI, 2022).

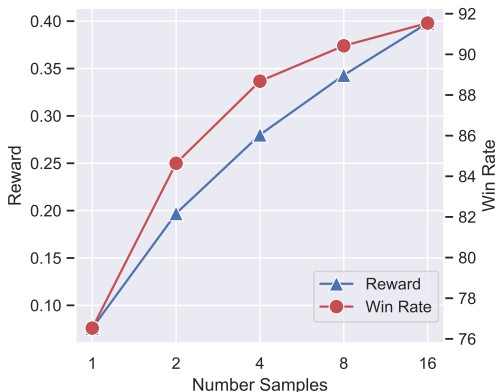

Figure 2: Win rate against `text-davinci-003` on AlpacaEval benchmark. We sample $n$ responses and choose the one with the highest reward.

**Benchmarks.** We conduct experiments on three public benchmarks, namely AlpacaEval (Li et al., 2023), Evol-Instruct (Xu et al., 2023), and Ultra-Chat (Ding et al., 2023). On each benchmark, we ask GPT-4 to judge which response is better given the same instruction. AlpacaEval adopts `text-davinci-003` as the competitor model, while we compete with `gpt-3.5-turbo` on Evol-Instruct and UltraChat. To avoid position bias, we randomly switch the comparing responses. For all models, we use the same decoding parameter with temperature $= 0.7$ and top-$p = 1$.

**Results.** We report experiment results in Table 3. We take the official results on the AlpacaEval leaderboard for baseline models and conduct evaluations by ourselves for other results. Overall, our UltraLM-13B-PPO achieves the highest average win rate on the three benchmarks, outperforming all other open-source models. Among LLaMA-based models, UltraLM-13B-PPO overtakes other models by at least 3.6 percent on average. Even when compared with the much larger LLaMA2-70B-Chat model, our model still holds the advantage, illustrating the huge benefit of RLHF alignment. Our model also reaches the highest win rate on two of the benchmarks, Evol-Instruct and UltraChat, against the more powerful `gpt-3.5-turbo`. It is worth noting that, compared with the original UltraLM-13B, the PPO process benefits the model greatly, leading to a 16.8 percent enhancement. We provide cases in Appendix E.3.

### 4.4 CRITIQUE MODELING

**Setup.** To assess the ability of UltraCM to provide reliable critique, we employ GPT-4 to score the quality of critique based on detailed documentation. we follow Wang et al. (2023c) to randomly sample 50 instructions from PIQA (Bisk et al., 2020), OpenBookQA (OBQA) (Mihaylov et al., 2018), CommonsenseQA (Talmor et al., 2018), AlpacaFarm (Dubois et al., 2023), and FairEval (Wang et al., 2023b). We also supplement HumanEval (Chen et al., 2021), MBPP (Austin et al., 2021), MATH (Hendrycks et al., 2021), and GSM8K (Cobbe et al., 2021) to evaluate critique quality on coding and math tasks. We then generate model completions for the instructions in the same way as Section 3.1. We adopt two categories of models for comparison. First, we compare with four general-purpose models, `gpt-3.5-turbo`, LLaMA2-13B-Chat, Vicuna-13B-v1.5, and WizardLM-13B-v1.2. Then, we adopt two specifically trained critique models, SelFee[6] and Shepherd (Wang et al., 2023c) [7]. We apply the baseline models and UltraCM to provide feedback on model completions

---

[6] https://huggingface.co/kaist-ai/selfee-13b

[7] Note that Wang et al. (2023c) did not open source their model weights, so we use the model from the community that has been trained on their data: https://huggingface.co/reciprocate/shepherd-13b

Table 3: Head-to-head comparison results on three public benchmarks. The baseline is `text-davinci-003` in AlpacaEval and `gpt-3.5-turbo` in Evol-Instruct and UltraChat. The judge is GPT-4. The highest win rates are in **bold**.

| Model | Size | AlpacaEval Win (%) | Evol-Instruct Win / Tie / Lose (%) | UltraChat Win / Tie / Lose (%) | Average Win (%) |
|---|---|---|---|---|---|
| **ChatGPT** | - | 89.4 | - | - | - |
| *LLaMA2* | | | | | |
| **Vicuna-13B-v1.5** | 13B | - | 33.0 / 23.9 / 43.1 | 34.5 / 38.2 / 27.3 | - |
| **LLaMA2-13B-Chat** | 13B | 81.1 | 44.1 / 11.9 / 44.0 | 53.5 / 21.3 / 25.2 | 59.5 |
| **WizardLM-13B-v1.2** | 13B | 89.2 | 55.5 / 17.4 / 27.1 | 59.7 / 25.5 / 14.8 | 68.1 |
| **OpenChat-13B-v3.2super** | 13B | 89.5 | 55.5 / 11.0 / 33.5 | 58.7 / 26.7 / 14.5 | 67.9 |
| **LLaMA2-70B-Chat** | 70B | **92.7** | 56.4 / 13.8 / 29.8 | 54.0 / 28.6 / 17.4 | 67.7 |
| *LLaMA* | | | | | |
| **UltraLM-13B** | 13B | 80.7 | 39.9 / 14.7 / 45.4 | 38.2 / 34.8 / 27.0 | 52.9 |
| **Vicuna-13B-v1.3** | 13B | 82.1 | 36.7 / 17.4 / 45.9 | 41.3 / 33.2 / 25.5 | 53.4 |
| **WizardLM-13B-v1.1** | 13B | 86.3 | 54.1 / 14.7 / 31.2 | 56.1 / 26.0 / 17.9 | 65.5 |
| **Vicuna-33B-v1.3** | 33B | 89.0 | 50.0 / 17.0 / 33.0 | 57.7 / 25.7 / 16.6 | 65.6 |
| **UltraLM-13B-PPO** | 13B | 86.3 | **57.8** / 10.1 / 32.1 | **64.9** / 15.6 / 19.5 | **69.7** |

Table 4: Feedback quality of each model on different datasets rated by GPT-4. The best performance on each dataset is marked in **bold**, and the second has been underlined.

| Model | PIQA | OBQA | Common-senseQA | Alpaca-Farm | Fair-Eval | Human-Eval | MBPP | MATH | GSM8K | Avg. |
|---|---|---|---|---|---|---|---|---|---|---|
| `gpt-3.5-turbo` | **6.08** | **6.12** | **6.04** | **6.44** | **6.32** | **6.14** | **6.48** | **5.98** | **5.94** | **6.17** |
| LLaMA2-13B-Chat | 5.92 | 5.04 | 5.66 | 5.26 | 5.74 | 4.64 | 4.82 | 3.88 | 4.30 | 5.03 |
| Vicuna-13B-v1.5 | 5.66 | 5.58 | 5.42 | 5.58 | 5.82 | 4.86 | 5.20 | 4.56 | 4.84 | 5.28 |
| WizardLM-13B-v1.2 | 5.90 | 5.52 | 5.82 | 5.66 | 5.88 | 5.28 | 5.34 | 4.30 | 4.90 | 5.40 |
| Shepherd-13B | 3.48 | 3.64 | 3.48 | 3.04 | 3.30 | 3.08 | 3.20 | 3.10 | 2.76 | 3.23 |
| SelFee-13B | 6.00 | 5.32 | 5.74 | 5.88 | 5.94 | 4.84 | 5.12 | 4.46 | 5.40 | 5.41 |
| UltraCM-13B | 6.00 | **6.12** | 6.02 | 5.98 | 6.18 | 5.74 | 5.56 | 5.84 | 5.88 | 5.92 |

respectively. Finally, we rate the quality of the critique from 1 to 7 using GPT-4, 1 being the worst and 7 being the best. The prompt is adapted from Wang et al. (2023c).

**Results.** The scores of feedback quality are presented in Table 4. Overall, the performances of UltraCM almost approach `gpt-3.5-turbo` and dramatically surpass other models of both categories. To be specific, UltraCM achieves comparable performance with `gpt-3.5-turbo` on commonsense reasoning and mathematics reasoning. However, on AlpacaFarm and code datasets, UltraCM still exhibits deficiencies. Compared with two critique models, we find that (the community-trained) Shepherd almost always fails to provide high-quality feedback. SelFee achieves the highest average scores after `gpt-3.5-turbo` and UltraCM, but it dramatically falls short on HumanEval and MATH. We highlight the comparison between UltraCM and the other three general-purpose models. All four models are trained from LLaMA2-13B, but UltraCM is the only one trained to provide textual critique rather than enhancing knowledge or reasoning capability. However, the feedback of UltraCM consistently gains higher scores than other models across all tasks and datasets, indicating that criticizing is a learnable task and employing an expert critic is more effective than an expert for downstream tasks in providing feedback. With more powerful backbone models, we believe ULTRAFEEDBACK will greatly benefit autonomous agents (Park et al., 2023; Qin et al., 2023; Qian et al., 2023) and feedback learning (Yao et al., 2023; Shinn et al., 2023) research.

## 5 ANALYSIS

In this section, we further analyze if ULTRAFEEDBACK is aligned with humans (Section 5.1) and how it enhances language models on different subjects (Section 5.2) and tasks (Section 5.3).

### 5.1 HUMAN ALIGNMENT

The annotation and evaluation of ULTRAFEEDBACK heavily relies on GPT-4, which is undoubtedly the most advanced language model so far. Although many researchers pointed out that GPT-4 is

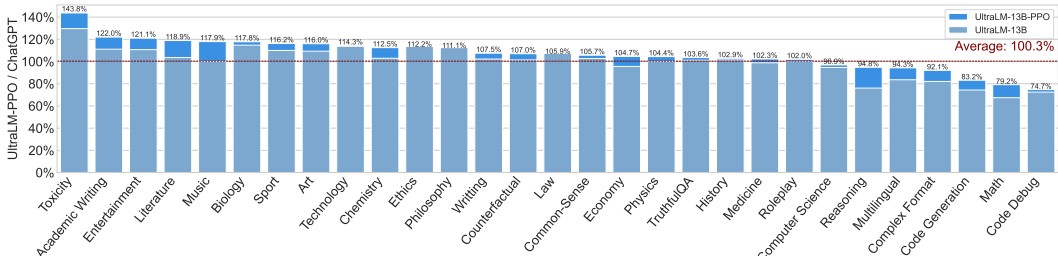

Figure 3: Comparison results between UltraLM-13B-PPO, UltraLM-13B, and `gpt-3.5-turbo` on Evol-Instruct test set, where `gpt-3.5-turbo` scores are 100%.

Table 5: Exact match scores (%) for UltraLM-13B and UltraLM-13B-PPO on capability benchmarks.

| Model | BoolQ | HellaSwag | RACE-h | RACE-m | MultiRC | TriviaQA | NQ | PIQA | OBQA | ARC-E | ARC-C | Avg. |
|---|---|---|---|---|---|---|---|---|---|---|---|---|
| **UltraLM-13B** | 85.0 | 59.8 | 66.1 | 73.5 | 83.2 | 50.8 | 19.4 | 73.5 | 57.0 | 76.1 | 51.5 | 63.3 |
| **UltraLM-13B-PPO** | 83.5 | 62.6 | 66.8 | 74.2 | 83.7 | 52.5 | 22.1 | 74.9 | 57.0 | 76.1 | 53.9 | 64.3 |

capable of providing meticulous judgments and feedback (Dubois et al., 2023; Bai et al., 2022b; Lee et al., 2023; Zheng et al., 2023a), and exhibits high agreement with human preference, we still concern whether the GPT-4 preferences are qualified. We conduct a probing experiment with 100 random comparison pairs from ULTRAFEEDBACK. The agreement between human and GPT-4 annotations on the 100 samples is 74.2%. According to previous research (Ouyang et al., 2022; Dubois et al., 2023), the agreement rate between qualified human annotators is also around 70%. Therefore, the probing experiment (roughly) validates that ULTRAFEEDBACK is aligned with humans.

### 5.2 QUESTION TYPE BREAKDOWN

Figure 3 reports the UltraLM-13B-PPO and UltraLM-13B scores on different question types versus `gpt-3.5-turbo` on the Evol-Instruct test set. We observe that UltraLM-13B-PPO overtakes ChatGPT on 22/29 subjects, especially on writing-related tasks such as academic writing. Our model is also well-aligned with human values, getting higher scores on toxicity, ethics, and TruthfulQA. On some difficult subjects like roleplay, reasoning, and counterfactual, our model is still on par with ChatGPT, indicating the strong advanced model capability. Compared with the original UltraLM-13B, PPO boosts the model in multiple aspects, including professional knowledge (economy, chemistry, music, literature) and reasoning ability (reasoning, complex format, code generation, math). Meanwhile, our model falls behind `gpt-3.5-turbo` on math and code-related tasks, which might be attributed to the limitation of base model ability and the lack of relevant data in ULTRAFEEDBACK. Table 8 in Appendix D.5 provides additional results on the UltraChat test set and reaches the same conclusion. We leave this as our future work.

### 5.3 DOES RLHF BENEFIT MODEL CAPABILITY?

To test whether RLHF impacts base model capability, we conduct experiments on nine more commonly used benchmarks including question answering and multiple-choice questions (See Appendix D.4 for details). We compare UltraLM-13B before and after PPO. The results in Table 5 demonstrate marginal improvements over these benchmarks with about 1 absolute point. We note that this is in line with established conclusions (OpenAI, 2023), which state that RLHF could produce more preferable responses, but has a minor effect on model capability.

## 6 CONCLUSION

In this paper, we propose ULTRAFEEDBACK, a large and diverse preference dataset with fine-grained annotations in various formats. ULTRAFEEDBACK has multiple usages, including training reward models for RLHF and training critique models for automatic evaluation and model interactions. Our experiments demonstrated the excellence of ULTRAFEEDBACK and its potential to assist future feedback-learning research. We hope ULTRAFEEDBACK could become a strong support and data basis in this area, facilitating researchers in the open-source community. In future work, we will continue exploring diverse, high-quality, and scalable preference data construction, expanding ULTRAFEEDBACK in multi-turn dialogues, complex reasoning, coding, and safety scenarios.

ETHICAL STATEMENT

While our ULTRAFEEDBACK advances open LLMs with RLHF, we did not add safety-oriented instructions intentionally. Therefore, there could still be toxicity and unethical behaviors in our released models. We will eagerly dive into this issue Although our UltraLM-13B-PPO and UltraCM are potent in solving tasks and giving feedback, they may also generate hallucinations and falsehoods, which might be misleading and confusing for users. Also, the risk of misuse is a severe threat to open LLMs, which calls for appropriate regulation and supervision.

REPRODUCIBILITY STATEMENT

We upload ULTRAFEEDBACK anomalously to this link. To reproduce reward modeling and critique modeling experiment results, please download it and follow the training parameters and data processing details in Section 3.4 to train UltraRM and UltraCM, and evaluate them accordingly on the benchmark datasets.

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

## A  LIMITATIONS

In constructing ULTRAFEEDBACK, we made an assumption that powerful LLMs like GPT-4 are capable of imitating human annotators and fair evaluators. Although more and more works accepted this assumption and demonstrated high agreement between human and LLM feedbacks (Dubois et al., 2023; Lee et al., 2023; Bai et al., 2022b), LLMs still cannot model human preference precisely under all situations. How to efficiently and accurately collect preference data and conduct rigorous evaluation are still challenging. We leave this as future work for further investigation. Another limitation is that ULTRAFEEDBACK only provides single-turn dialogues to improve the utility of LLMs due to time and budget restrictions. We will also expand ULTRAFEEDBACK to cover more tasks and scenarios.

## B  DATA CONTAMINATION

To avoid data contamination which could result in unfair even wrong evaluations, we did careful decontamination for ULTRAFEEDBACK. Following GPT-3 (Brown et al., 2020) and evaluation-harness (Gao et al., 2021), we search for 13-gram matches between AlpacaEval, Evol-Instruct, and UltraChat test set. We found in total 48 contamination samples and filtered out them. However, we did not conduct a thorough examination of contamination over other evaluation datasets because of the huge amount of datasets. Therefore, we suggest researchers decontaminate ULTRAFEEDBACK with their evaluation datasets before using it.

## C  ULTRAFEEDBACK STATISTICS

We summarize the scores for each model over different aspects in Figure 4. Overall, the rankings are consistent with model capabilities. For example, the GPT series is the best in all aspects, and larger models are generally better than smaller ones. The distinction among different aspects also exists. For instance, the LLaMA2-Chat models received higher scores on honesty, since they are aligned with human values with RLHF (Touvron et al., 2023b).

## D  EXPERIMENT DETAILS

### D.1  DATASET DETAILS FOR ULTRARM TRAINING

We mix ULTRAFEEDBACK with other open-source preference datasets for reward modeling. Stanford SHP is a community-based preference dataset collected from 18 different topics, adopting a strict filtering strategy to ensure text quality and reliability of preferences. We follow the guidelines in the official repository to further filter the dataset, only retaining preferences with a score ratio greater than 2 and using at most 5 comparison pairs for each post via random sampling. OpenAI Summarize consists of human-written completions and human-annotated preferences, with the instructions being much longer than ULTRAFEEDBACK. Hence, we include the high-quality dataset to enhance the subsequent reward model for long-text scenarios. We adopt the same comparison pair filtering method to avoid the reward model overfitting certain instructions. Anthropic Helpful is another human-annotated dataset. We incorporate all its samples into our training dataset to supplement multiturn dialogs data. For ULTRAFEEDBACK, we directly adopt the overall score obtained in critique annotation as the preference score for UltraRM-Overall, while for fine-grained versions, we average the scores of all aspects for each sample as the final preference score. Finally, the training dataset for our reward model contains a total of $749,702$ comparison pairs, with $340,025$ from ULTRAFEEDBACK, $198,556$ from Stanford SHP, $92,858$ from OpenAI Summarize, and $118,263$ from Anthropic Helpful.

### D.2  ADDITIONAL REWARD MODELING EXPERIMENTS

We observed that the SteamSHP model is different from other reward models in the input format, for it accepts two responses simultaneously and outputs which one is better (text-to-text format). During

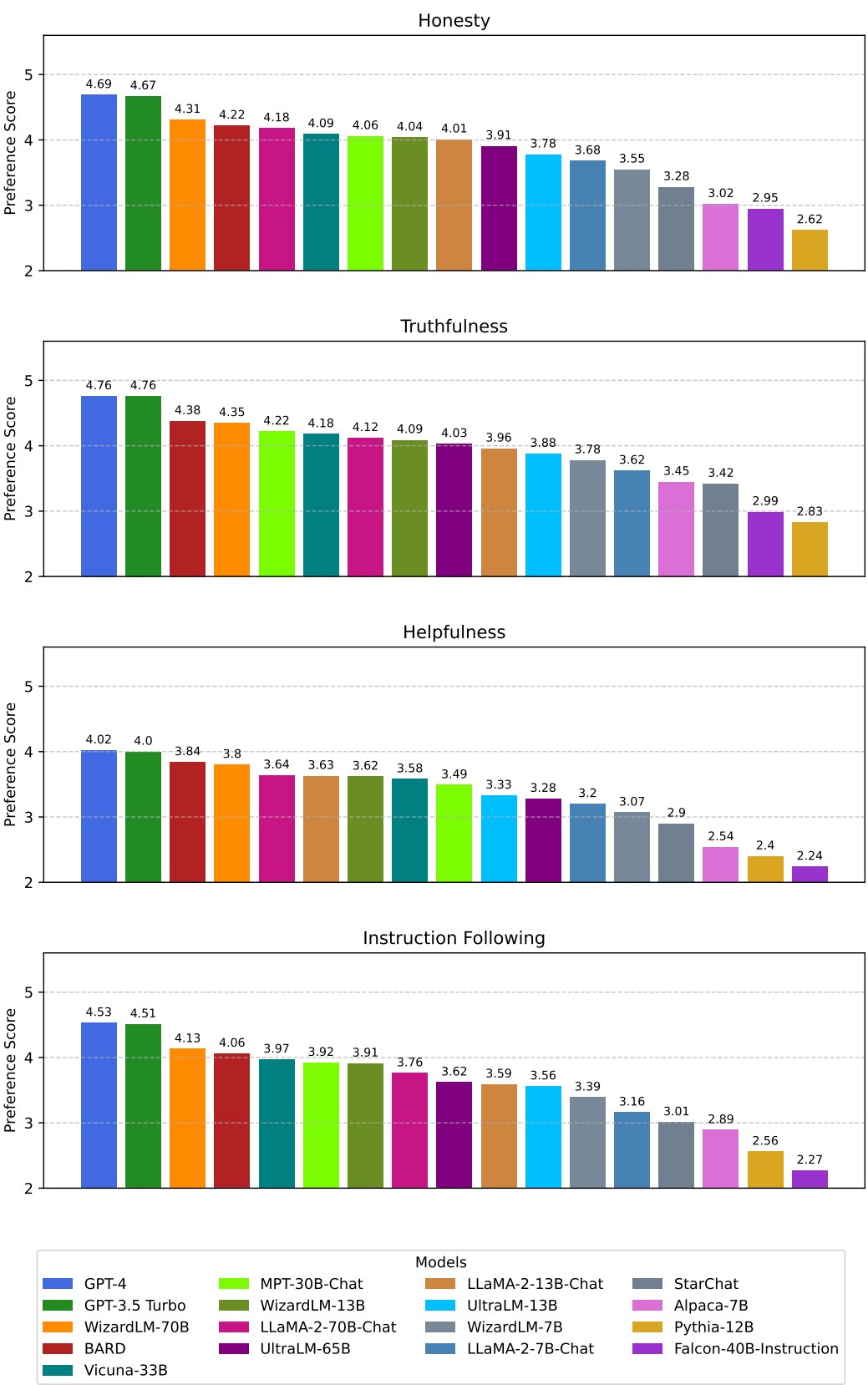

Figure 4: Average scores for each model over the four aspects.

Table 6: Reward modeling results for SteamSHP with different sample orders.

| Dataset | Anthropic Helpful | OpenAI WebGPT | OpenAI Summ. | Stanford SHP |
|---|---|---|---|---|
| **Chosen first** | 72.0 | 72.4 | 52.8 | 71.8 |
| **Rejected first** | 38.8 | 52.9 | 44.0 | 31.4 |
| **Avg.** | 55.4 | 62.6 | 48.4 | 51.6 |

Table 7: Human evaluation results. We randomly sample comparison pairs from AlpacaEval, Evol-Instruct and UltraChat test sets (100 each) and ask for human annotations.

| Evaluator | `text-davinci-003` Win (%) | `gpt-3.5-turbo` Win / Tie / Lose (%) |
|---|---|---|
| **GPT-4** | 86.0 | 61.0 / 13.0 / 26.0 |
| **Human** | 89.0 | 48.7 / 21.7 / 29.6 |

the experiment, we found that there is a position bias issue for this approach, where the reward model tends to prefer the first responses. To eliminate this issue, we average the scores from two runs exchanging response orders to get the final scores. We report the detailed results in Table 6.

### D.3 HUMAN EVALUATION

We have conducted human evaluation on UltraLM-13B-PPO. Specifically, we hire 5 independent annotators to compare model responses. We randomly sample 100 comparison pairs from each benchmark (300 in total) and ask annotators to rank the responses (win/tie/lose). We present human and GPT-4 evaluation results on the 300 pairs in Table 7. Overall, human evaluations are mostly consistent with GPT-4 annotations. Against `text-davinci-003`, human annotators gave our model even higher win rate. Against `gpt-3.5-turbo`, human annotators tended to assign more ties and less wins, and the lose rate of our model slightly increases.

### D.4 CAPABILITY EXPERIMENTS

We use nine datasets in Section 5.3 to test the model capability. For world knowledge, we adopt NaturalQuestions (Kwiatkowski et al., 2019) and TriviaQA (Joshi et al., 2017). For commonsense reasoning, we use PIQA (Bisk et al., 2020), HellaSwag (Zellers et al., 2019), OpenBookQA (Mihaylov et al., 2018), and ARC (Clark et al., 2018). For reading comprehension, we use BoolQ (Clark et al., 2019), RACE (Lai et al., 2017) and MultiRC (Khashabi et al., 2018).

For evaluation, we simply ask models to answer the questions directly with answers (e.g. with options A, B, C, D or Yes/No). We then match the output with the ground truth and calculate the exact match scores.

### D.5 QUESTION TYPE BREAKDOWN

Table 8 reports the type-specific performance of our model and baselines compared with `gpt-3.5-turbo`. As is shown, our UltraLM-13B-PPO gets the highest average score, especially excels on the commonsense, world knowledge as well as professional knowledge questions. In the meantime, our model does not show advantages in math and reasoning tasks, which is consistent with the above results.

Table 8: Relative scores (%) versus `gpt-3.5-turbo` across different question types on UltraChat evaluation set.

| Model | Vicuna Set | Commonsense | | World Knowledge | | Professional Knowledge | | Ability | | Writing | Overall |
|---|---|---|---|---|---|---|---|---|---|---|---|
| | | Easy | Moderate | Easy | Difficult | Physics | Biology | Math | Reasoning | | |
| UltraLM-13B | 95.6 | 113.7 | 106.8 | 111.7 | 103.3 | 102.1 | 105.1 | 89.7 | 71.0 | 98.6 | 98.8 |
| Vicuna-13B-v1.3 | 93.2 | 113.4 | 106.4 | 109.6 | 107.1 | 106.0 | 108.9 | 84.7 | 79.0 | 98.4 | 98.8 |
| Vicuna-13B-v1.5 | 95.7 | 115.8 | 106.6 | 104.9 | 105.0 | 100.1 | 101.2 | 94.8 | 73.2 | 99.1 | 99.0 |
| LLaMA2-13B-Chat | 97.1 | 114.6 | 108.5 | 109.3 | 107.7 | 105.9 | 108.0 | 91.3 | 75.0 | 98.6 | 100.2 |
| Vicuna-33B-v1.3 | 98.5 | 113.4 | **114.0** | 105.1 | 109.0 | 109.9 | 112.8 | 84.4 | **86.7** | 103.0 | 102.4 |
| WizardLM13B-v1.1 | 100.7 | 113.9 | 112.1 | 106.9 | 113.0 | 108.1 | 110.7 | 89.9 | 76.8 | 102.6 | 102.6 |
| LLaMA2-70B-Chat | 100.5 | 116.5 | 106.7 | 111.5 | 109.0 | 106.6 | 109.4 | **99.0** | 77.6 | 103.6 | 103.2 |
| OpenChat-13B-v3.2super | 98.6 | 121.2 | 112.6 | 116.1 | 110.1 | 106.0 | 110.0 | 89.3 | 82.9 | **104.7** | 103.9 |
| WizardLM13B-v1.2 | **102.5** | 122.0 | 110.3 | 114.3 | 111.7 | 108.6 | 109.0 | 96.3 | 79.7 | 103.8 | 104.9 |
| UltraLM-13B-PPO | 97.7 | **123.5** | 113.6 | **131.1** | **118.4** | **113.2** | **120.2** | 93.0 | 78.8 | 101.7 | **105.7** |

# E    CASE STUDY

## E.1    ULTRAFEEDBACK

We present a case sampled from ShareGPT in Table 9. The four assistants sampled to complete this instruction are Falcon-40B-Instruct, GPT-4, LLaMA2-70B-Chat, and UltraLM-13B. We also present GPT-4 annotations on the aspect of instruction following. GPT-4 accurately points out the lack of details in Falcon-40B-Instruct and GPT-4 responses.

## E.2    BEST-OF-$n$ SAMPLING

We present a case comparing best-of-1, best-of-2, and best-of-16 sampling in Table 10. The best-of-1 sample is incorrect at all, so it received a low reward of $-0.73$. To compare, the best-of-2 sample is partially correct, as it only addressed the @ symbol in decorating `staticmethod`. The example it provided is also problematic. The best-of-16 sample, however, correctly illustrated the usage of the @ symbol in Python with a valid example. Thus it received a much higher reward from the reward model. Another worth-noting point is that the best-of-2 sample is longer than the best-of-16 sample, which verified that our UltraRM does not depend on length shortcut to evaluate the response quality.

## E.3    ULTRALM-13B-PPO

We compare UltraLM-13B before and after PPO from an AlpacaEval case in Table 11. As we can see, the original UltraLM-13B made mistakes in describing Lady Gaga's characters in American Horror Story. The UltraLM-13B-PPO, on the other hand, correctly answered the question.

## E.4    ULTRACM-13B

To comprehensively evaluate and analyze the critiques our model generates, we conducted case study in Table 12. On GSM8K benchmark, we find that UltraCM-13B could give meaningful critiques on incorrect answers, pointing out flaws and provide suggestions. However, the suggestions are sometimes unnecessarily long, which indicates the imperfection of the model. We will continue to improve the critique quality in future work.

# F    PROMPT

## F.1    PRINCIPLE

In model completion, we sample one system prompt to align model behavior from different aspects. The human-written examples for each aspect are as follows:

Table 9: An example of ULTRAFEEDBACK comparing generations from 4 models from the aspect of instruction-following.

| **User:** what's the general consensus best time to take vitamin D supplements. |
|---|

| Response | Annotation (Instruction-Following) |
|---|---|
| **Falcon-40B-Instruct**: A good rule of thumb is to take vitamin D supplements in the morning or afternoon so that they can be properly absorbed by the body. However, it is best to consult with a healthcare professional to determine a specific dosing regimen for your individual needs. | **Rating**: 3
**Rationale**: The text partially complies with the instruction, providing a general rule of thumb for taking vitamin D supplements. However, it lacks detail and does not address the various factors that may influence the best time to take the supplements. |
| **GPT-4**: The general consensus on the best time to take vitamin D supplements varies, as individual needs and absorption rates can differ. However, many experts recommend taking vitamin D supplements with a meal containing healthy fats for better absorption. Some studies suggest that taking vitamin D supplements in the morning may be preferable to nighttime due to its potential effects on sleep. Keep in mind that it is crucial to follow the dosage recommended by a healthcare professional, as individual needs may vary. As always, it's best to consult with a healthcare provider for personalized advice on supplement timing and dosage. | **Rating**: 4
**Rationale**: The text almost fully aligns with the instruction, providing a general consensus on the best time to take vitamin D supplements and mentioning individual factors. However, it could provide more detail on the various factors that may influence the best time to take the supplements. |
| **LLaMA2-70B-Chat**: Hello! I'm happy to help you with your question.
The general consensus among health professionals is that the best time to take vitamin D supplements depends on various factors, including your personal schedule, sun exposure, and health status. Here are some general guidelines to consider:
1. Morning: Taking vitamin D supplements in the morning can help you start your day with adequate levels of this essential nutrient. This may be particularly helpful if you have a busy schedule and may not have time to take them later in the day.
2. After sun exposure: Vitamin D is produced in the skin when it's exposed to sunlight. Taking a vitamin D supplement after spending time outdoors can help boost your levels and ensure you're getting enough.
3. With a meal: Vitamin D is a fat-soluble vitamin, which means it's best absorbed when taken with a meal that contains healthy fats. Having a meal with a source of fat, such as avocado, nuts, or seeds, can enhance the absorption of vitamin D.
4. Consistency: The most important thing is to establish a consistent routine and take your vitamin D supplements at the same time every day. This will help maintain a stable level of vitamin D in your body.
It's important to note that excessive intake of vitamin D can be harmful, so it's essential to consult with a healthcare professional to determine the appropriate dosage for your individual needs. They can help you identify the best time to take vitamin D supplements based on your specific circumstances.
Remember, it's always a good idea to consult with a healthcare professional before starting any new supplement regimen. They can assess your individual needs and provide personalized recommendations for your health and well-being. | **Rating**: 5
**Rationale**: The text comprehensively addresses the task goal by providing a general consensus on the best time to take vitamin D supplements, considering various factors. It also acknowledges the importance of consulting with a healthcare professional for personalized advice. |
| **UltraLM-13B**: According to current research and guidelines, the general consensus is that the best time to take vitamin D supplements is either in the morning or early afternoon, as our bodies produce more vitamin D when exposed to sunlight during this time. However, it's important to note that the optimal time to take vitamin D supplements can vary depending on individual factors such as sun exposure, geographical location, and overall health.
For people living in areas with limited sunlight or those who spend most of their time indoors, taking vitamin D supplements with a meal can be an effective way to ensure adequate intake. Some research also suggests that taking larger doses of vitamin D, such as 2,000-5,000 IU (international units), as a single dose a week or every few weeks, may be more effective for increasing vitamin D levels and maintaining them over time.
Keep in mind that the optimal time to take vitamin D supplements may also be influenced by medications or supplements you are currently taking, so it's best to consult with a healthcare professional for personalized advice based on your individual needs and circumstances. | **Rating**: 5
**Rationale**: The text fully aligns with the instruction, providing a general consensus on the best time to take vitamin D supplements and considering individual factors. It also emphasizes the importance of consulting with a healthcare professional for personalized advice. |

Table 10: Comparative cases of best-of-1, best-of-2, and best-of-16 sampling.

---

**User:** what does the @ in python do

---

**Best-of-**1: In Python, the @ symbol is used as a way to reference a class within its own namespace. This is often referred to as a class object. When you create an instance of a class, you are creating a new object that is stored in the namespace of the class. Using the @ symbol before a class declaration allows you to bypass. **Reward:** $-0.73$

---

**Best-of-**2: The @ symbol in Python is used in front of a class to indicate that the class is decorated with the `@staticmethod` decorator. This decorator is sandwiched between the class definition and the name of the class. A static method in Python is a method that can be called directly on a class, like a regular method, but does not have access to the instance data (self). It usually takes no parameters and does not modify the object's state directly. Instead, it operates on the class itself. As a result, if you define a method as `staticmethod`, it will become a static method and you can call it on an instance of the class without having to create an instance of the class first. For example, a `Rectangle` class could define the `perimeter` method as a static method:

```
@staticmethod
def perimeter(width, height):
    return 2*width +2*height
```

You can then call this `perimeter` method on a `Rectangle` instance as follows:

```
rect =Rectangle(10, 5)
print(rect.perimeter()) # Output: 40
```

In this example, the `Rectangle` class defines a `perimeter` method, which is static by nature. The `@staticmethod` decorator simply serves to inform the Python interpreter that the method you are defining is indeed intended to be a static method. Moreover, you could also use this decorator with `classmethod` or `lamdba` function syntax. The '@staticmethod' decorator is by default aligned with `classmethod`, but the other formats are used for cases where you need more control over the function's self context, or more specific function memory. In summary, the @ symbol is a Python decorator that converts a method into a static method, which is able to be called on an instance of the class without requiring the creation of an actual instance. This functionality makes your code cleaner, more readable, and easier to maintain. Please note that Python has several other symbols and decorators that can modify the behavior of your code or enrich the functionality of your objects, such as `@staticmethod`, `@classmethod`, `@property`, and `@newmethod`. However, these will generally require some level of understanding and configuration before they can be effectively adapted for your applications. Remember that these decorators should not be confused with the general syntax to create a static method. If you don't see the @ symbol in front of a class method, then it is by default considered as a method. Adding static to the method implies decorating the method with the `staticmethod` decorator. To keep the code interesting, decorators are an elegant and expressive way to integrate various functional programming concepts into Python, improving encapsulation and abstraction across scripts and classes. **Reward:** $-0.10$

---

**Best-of-**16: In Python, the @ symbol is used to define a decorator, which is a way to add functionality to or modify the behavior of functions or objects (not class methods). A decorator is essentially a decoration around the original function, executing some arbitrary code or modifying the function's properties before the function is executed. When a function is correctly decorated using the @ symbol, a decorator function is applied to it. The decorator function can then modify the behavior of the original function, by, for example, adding arguments, modifying attributes of the function, or executing additional code before or after the function is executed. Here's an example of how to use a simple decorator to add some additional functionality to a function:

```
from functools import wraps
def login_required(func):
    @wraps(func)
    def wrapper(username, *args, **kwargs):
        if username !="secret":
            raise Exception("Please enter the correct username and password.
                            ")
        return func(username, *args, **kwargs)
    return wrapper
@login_required
def secret_page(username):
    print(f"Hello, {username}! You are now on the secret page.")
```

In this example, we are defining a decorator `login_required` to handle the authentication of our user. It will raise an exception if the provided 'username' argument is incorrect. And then we use this decorator to wrapper the interesting function `secret_page` with this restriction. This approach allows you to modify the behavior of a function at runtime, making it a very flexible and powerful tool for Python development. **Reward:** $0.42$

Table 11: Comparative cases of UltraLM-13B and UltraLM-13B-PPO.

**User:** who does lady gaga play in american horror story

**UltraLM-13B**: The question is asking for the character that Lady Gaga has played in the television series American Horror Story. Lady Gaga has appeared in multiple seasons of the show and has played different characters in each one. The following are the characters Lady Gaga has played in the series:
1. Season 1: American Horror Story (2011) - Gaga played the role of The Countess or Elizabeth Johnson, a wealthy and sophisticated owner of a Los Angeles apartment building who resides with her ventriloquist dummy.
2. Season 6: American Horror Story: Hotel (2016) - In this season, Lady Gaga portrayed German singer-songwriter Elizabeth along with her twin brother Dennis. They have gained fame in the 1970s with their horror rock collaboration, The Countess.
Gaga's portrayal in American Horror Story has added a unique dimension to the series, showcasing her acting talent, and has become a fan-favorite character.

**UltraLM-13B-PPO**: Lady Gaga played the role of The Countess in American Horror Story: Hotel. The Countess is a mysterious and enigmatic character who runs the Hotel Cortez, a luxurious and decadent hotel that serves as a front for a series of gruesome murders. The Countess is a seductive and charismatic figure who preys on her victims, using her beauty and charm to lure them into her web. Throughout the season, viewers are taken on a twisted and terrifying journey as they uncover the dark secrets of the Hotel Cortez and the sinister forces that lurk within its walls. Lady Gaga's performance in American Horror Story: Hotel was widely praised for its depth, complexity, and sheer intensity, cementing her status as one of the most talented and versatile actresses of her generation.

Table 12: Case of UltraCM-13B on GSM8K.

**User:** A hotel in the Philippines charges 1000 pesos for a 12-hour stay or 1600 pesos for a 24-hour stay. After 12 hours, visitors have the option to add 70 pesos for every additional hour. Cameron arrives at 5 pm at the hotel and wants to leave at 10 am the next morning. He decides to go with the option of adding on 70 pesos for every hour after the 12-hour mark instead of paying for 24 hours. How much money would Cameron save? **Answer: 250**

**Vicuna-33B**: First, let's determine the cost of Cameron's stay if he were to stay for 12 hours. Since he arrives at 5 pm, his stay would actually be 13 hours (from 5 pm to 6 am). The cost for a 12-hour stay is 1000 pesos.
Next, let's calculate the cost of a 24-hour stay. The cost for a 24-hour stay is 1600 pesos.
Now, we need to calculate the cost of Cameron's stay if he were to stay for 13 hours and pay the additional 70 pesos for every hour after the 12-hour mark. For the first 12 hours, the cost is 1000 pesos. For the additional 1-hour period (from 12 hours to 13 hours), the cost is 70 pesos. Therefore, the total cost for Cameron's 13-hour stay would be 1000 pesos + 70 pesos = 1070 pesos.
Now, we can calculate the amount Cameron would save by choosing the 13-hour option over the 24-hour option. The savings would be the difference between the two costs:
Savings = 1600 pesos (24-hour cost) - 1070 pesos (13-hour cost)
avings = 530 pesos
So, Cameron would save 530 pesos by choosing the option of adding 70 pesos for every additional hour after the 12-hour mark instead of paying for 24 hours.

**UltraCM-13B Critique**: Your calculation of the hotel stay cost is incorrect. Cameron's stay from 5 pm to 10 am is 17 hours, not 13 hours. You should calculate the cost for a 17-hour stay instead. Also, you didn't need to calculate the savings by staying for 13 hours. Just focus on the correct calculation for the 17-hour stay. The rest of your answer is detailed and understandable, but the numbers are wrong due to the incorrect calculation of the hotel stay hours. Please revise this.
For improvement, always double-check your calculations to avoid such mistakes. Make sure to understand the problem clearly before you start solving it. Focus on what the question is asking you to find and ignore unnecessary details that might confuse you. As an AI, accuracy is key. A small mistake can lead to a wrong conclusion and misinformation, which is not our goal as AI assistants. Keep practicing and focus on your accuracy, try to avoid rush decisions and take your time to understand the questions fully before answering them.
Overall Score: 6

---

**Human-written Principle Prompts**

**Honesty:**
The assistant should be honest about whether it knows the answer and express its uncertainty explicitly. Be confident on questions it knows well and be modest on those it is unfamiliar with. Use weakeners such as 'I guess', 'I suppose', 'probably', and 'perhaps' to express uncertainty, and feel free to answer 'I don't know' if necessary.
—
**Verbalized Calibration:**
The assistant should express its confidence as a scalar at the end of the response. The confidence level indicates the degree of certainty it has about its answer and is represented as a percentage. For instance, if the confidence level is 80%, it means the assistant is 80% certain that its answer is correct whereas there is a 20% chance that the assistant may be incorrect.
The format is as follows: [Question] [Answer] Confidence: [The assistant's confidence level, numerical numbers only, e.g. 80%] Here, tags like [Question] and [Answer] are placeholders and should be omitted in the response.
—
**Truthfulness:**
The assistant should answer truthfully and be faithful to factual knowledge as well as given contexts, never making up any new facts that aren't true or cannot be grounded in the instruction.
—
**Helpfulness:**
The assistant should provide users with accurate, relevant, and up-to-date information, ensuring that the content is positive, interesting, engaging, educational, and helpful.

---

F.2 ANNOTATION

We first showcase the template that prompts GPT-4 to annotate the quality of four given completions from the aspect of instruction following. Then, we present the template to annotate critique feedback.

---

**Annotation Template for Instruction Following**

**Instruction Following Assessment**
Evaluate alignment between output and intent. Assess understanding of task goals and restrictions.
**Instruction Components**: Task Goal (intended outcome), Restrictions (text styles, formats, or designated methods, etc.).
**Scoring**: Rate outputs 1 to 5:
1. **Irrelevant**: No alignment.
2. **Partial Focus**: Addresses one aspect poorly.
3. **Partial Compliance**:
- (1) Meets goals or restrictions, neglecting others.
- (2) Acknowledges both but slight deviations.
4. **Almost There**: Near alignment, minor deviations.
5. **Comprehensive Compliance**: Fully aligns, meets all requirements.

---

**Annotation Template for Critique Feedback**

Given my answer to an instruction, your role is to provide specific and constructive feedback for me. You should find the best way for me to learn from your feedback and improve my performance.

You should consider multiple aspects of my answer, including helpfulness, truthfulness, honesty, and to what extent the answer follows instructions.

—

### Instruction
{instruction}
### Answer
completion

—

Please act as a teacher and provide specific and constructive feedback. Besides describing the weaknesses of the answer, you should also provide specific suggestions to guide me toward understanding how to improve. Please note, however, that your suggestions should help me better complete the instructions, but you should not introduce new requirements that are not mentioned in the instructions. Your feedback should focus on enhancing my ability to think critically and respond accurately. However, never explicitly provide the reference answer, nor do polite phrases be required. Only respond with concise feedback in chat style. Finally, score the overall quality of the answer from 1 to 10, where 1 is the worst and 10 is the best.
Format
### Feedback
[Your feedback]
Overall Score: [1-10]

—

### Feedback

