# OpenReview forum: "UltraFeedback: Boosting Language Models with High-quality Feedback"
_ICLR.cc/2024/Conference — Submitted to ICLR 2024_

### Official Review · Reviewer_cqtN · 2023-10-31

**Soundness:** 4 excellent
**Presentation:** 4 excellent
**Contribution:** 4 excellent
**Rating:** 8
**Confidence:** 3

**Summary:**

This paper proposes *UltraFeedback*, a large-scale, high-quality and diverse preference dataset for facilitating the research of RLHF. The dataset is constructed with the help of a set of instructions and models. The dataset will be open-sourced, which I think will be very helpful for the community. To verify the effectiveness of the dataset, the authors also provide a reward model UltraRM, and a critique model UltraCM.

**Strengths:**

1. The paper is well written and organized. It is easy to follow the dataset and methods proposed in the paper.

2. It is a common view in the era of LLMs that data scalability and quality are the key factors for training or instructing language models. *UltraFeedback* makes remarkable progress in the field by providing a high-quality and open-source preference dataset. It will greatly facilitate the further research of the community.

**Weaknesses:**

I did not see big problems in the paper.

But I have some minor concerns.

1. The annotations are heavily dependent on GPT4. Can we consider the models trained with the dataset as the distilled version of GPT4? Would the errors of GPT4 be propagated?

2. Are all the data annotated by GPT4? Is there a part of the data annotated by humans?

3. The instructions are all single turns? Does the dataset support the training of multi-turn scenarios?

**Questions:**

See the weakness part.

---

> ### Author Response · Authors · 2023-11-13
> **Response to Reviewer cqtN**
>
> Thanks for your appreciation! We will address your concerns as follows:
>
> > Q1: The annotations are heavily dependent on GPT4. Can we consider the models trained with the dataset as the distilled version of GPT4? Would the errors of GPT4 be propagated?
>
> A1: The models trained with UltraFeedback (e.g. UltraRM) distills the *preference* of GPT-4. This is truly a limitation of our work as discussed in Appendix A and we must admit that the most advanced GPT-4 may also make mistakes on annotations. However, during our preliminary experiments, we noticed that some tasks are difficult for human annotators, especially the ones with long inputs and/or outputs (e.g. translation, summarization). To compare, GPT-4 annotations are more scalable and efficient, and the quality is comparable with human annotators (please see Section 5.1 and [1,2,3]).
>
> > Q2: Are all the data annotated by GPT4? Is there a part of the data annotated by humans?
>
> A2: Yes, the released data are all annotated by GPT-4. We selected 100 samples for human annotation and discussed their alignment issue in Section 5.1.
>
> > Q3: The instructions are all single turns? Does the dataset support the training of multi-turn scenarios?
>
> A3: Yes, all instructions are single turns due to the budget limitation. To resolve this issue, we mixed UltraFeedback and other datasets like HH-RLHF and OASSIST1 to train UltraRM. As the experimental results indicate, UltraRM performs pretty good at multi-turn reward modeling on HH-RLHF.
>
>
>
> References
>
> [1] Dubois et al. Alpacafarm: A simulation framework for methods that learn from human feedback. 2023
>
> [2] Lee et al.  Rlaif: Scaling reinforcement learning from human feedback with ai feedback. 2023
>
> [3] Bat et al. Constitutional ai: Harmlessness from ai feedback. 2022

---

### Official Review · Reviewer_YWNs · 2023-11-01

**Soundness:** 2 fair
**Presentation:** 3 good
**Contribution:** 2 fair
**Rating:** 5
**Confidence:** 5

**Summary:**

This paper introduces ULTRAFEEDBACK, an expansive and varied dataset offering detailed annotations in multiple formats. This dataset is versatile and can be employed for tasks like training reward models in RLHF and creating critique models for automated assessments and model interactions.

**Strengths:**

The paper is well-written and easy to follow. It offers high quality data distilled from other LLMs and also a reward model for future research.

**Weaknesses:**

I'm unclear about the motivation the author introducing CRITIQUE MODELING in the paper. The scalar reward model is used for training PPO, it appears that CRITIQUE MODELING is solely for criticism? Is its purpose to distill the critique capabilities from larger models, e.g., ChatGPT? If so, I don't see how it relates to the scalar reward model. My primary interest lies in the interplay between the scalar reward model and PPO. How can we optimally utilize the reward model distilled from GPT-4? Moreover, how can we effectively leverage the four dimensions of preference data in both the reward model and PPO training?

The primary contribution of this paper appears to be sourcing query data from various LLMs, utilizing GPT-4 for ranking, and training with a standard reward model and PPO. I struggle to find significant innovation or contribution. While I concede that this paper may offer a valuable dataset for subsequent research, I have to say that the novelty in this study seems restricted. As I previously pointed out, the author maybe should concentrate on harnessing this potent reward model or the high-quality data to derive a more robust distilled model, akin to Zephyr.

**Questions:**

Please see the weakness.

---

> ### Author Response · Authors · 2023-11-17
> **Response to Reviewer YWNs**
>
> We appreciate your kind suggestions and feedback. Here are our responses:
>
> > Q1: The motivation of critique modeling
>
> A1: Sorry for the confusion. We view textual critiques as another valuable feedback format besides scalar rewards, which are
>
> - Human-friendly and Interpretable: Compared with scalar rewards, textual critiques are more readable and acceptable by humans. Also, critiques could be more flexible by pointing out flaws and providing suggestions.
>
> - Interactive: The generated critiques could be further integrated into the conversations with models. Some works also find that good critiques could further improve model performances [1,2,3].
>
>   To further study the properties of critiques, we conducted a case study in Appendix E.4. On math problem benchmarks, we found that UltraCM-13B could give meaningful critiques on incorrect answers, pointing out flaws and providing suggestions. However, the suggestions are sometimes unnecessarily long, which indicates the imperfection of the model. We will continue to improve the critique quality in future work.
>
> > Q2: The primary contribution of this paper appears to be sourcing query data from various LLMs, utilizing GPT-4 for ranking, and training with a standard reward model and PPO. I struggle to find significant innovation or contribution. While I concede that this paper may offer a valuable dataset for subsequent research, I have to say that the novelty in this study seems restricted. As I previously pointed out, the author maybe should concentrate on harnessing this potent reward model or the high-quality data to derive a more robust distilled model, akin to Zephyr.
>
> A2: Thanks for figuring this. Our main contributions, as described in the Introduction, are the high-quality preference dataset UltraFeedback along with it powered models UltraRM, UltraLM-PPO, and UltraCM. We believe these datasets and models are valuable for the open-source LLM community and are in line with the ICLR *datasets and benchmarks* track.
>
> On using the dataset, we conducted ablation studies on reward modeling (see Section 4.1) and validated the impact of data mixture and fine-grained reward signals. We also tested different use cases such as best-of-N sampling and PPO, and analyzed human alignment, improvement on various question types, and model capabilities. Due to the limitation of the computation budget, we were not able to do more experiments to further investigate the advanced usage of UltraFeedback, and it is out of the scope of our paper. We hope the open-source LLM community can find better utilization of UltraFeedback (like Zephyr).
>
> References
>
> [1] Wang et al. Shepherd: A critic for language model generation. 2023.
>
> [2] Ye et al. SelFee: Iterative Self-Revising LLM Empowered by Self-Feedback Generation. 2023.
>
> [3] Wang et al. MINT: Evaluating LLMs in Multi-turn Interaction with Tools and Language Feedback. 2023.

---

### Official Review · Reviewer_AWP7 · 2023-11-02

**Soundness:** 3 good
**Presentation:** 4 excellent
**Contribution:** 3 good
**Rating:** 6
**Confidence:** 3

**Summary:**

The paper proposes ULTRAFEEDBACK: A large-scale, high-quality, and diversified preference dataset for training and evaluating large language models (LLMs) with reinforcement learning from human feedback (RLHF).

**Data construction**: The paper describes how to sample diverse instructions and model responses from multiple sources, and how to use GPT-4 to provide fine-grained numerical and textual feedback for each response.

**Data characteristics**: The paper shows that ULTRAFEEDBACK is the largest, most diverse, and most fine-grained preference dataset in the open-source community, and that it can serve as both a preference and a critique dataset.

**Data applications**: The paper demonstrates how to use ULTRAFEEDBACK to train a state-of-the-art reward model (UltraRM) and a critique model (UltraCM) based on LLaMA2-13B, and how to use them to enhance open-source chat language models with best-of-n sampling and PPO.

**Strengths:**

- Originality: The proposed ULTRAFEEDBACK is a massive dataset of preferences and critiques for various natural language tasks, such as chat, summarization, translation, and more. This is the largest, most diverse, and most fine-grained preference dataset in the open-source community.
- Quality: The paper describes the data construction process in detail, explaining how to sample diverse instructions and responses from multiple sources, how to use GPT-4 to provide fine-grained numerical and textual feedback
- Clarity: The paper is well-written and easy to follow.
- Significance: The paper contributes a valuable resource of ULTRAFEEDBACK, which can serve as both a preference and a critique dataset for various natural language tasks. The paper also contributes novel models of UltraRM and UltraCM, which can learn from ULTRAFEEDBACK to provide rewards and critiques for any given model response. The paper demonstrates the practical impact of using UltraRM and UltraCM to enhance open-source chat language models with RLHF, showing that they can produce feedback more preferred by GPT-4.

**Weaknesses:**

- Lack of human evaluation: The paper uses GPT-4 for most of the evaluation. However, it would favor models (UltraCM, UltraLM-13B-PPO) trained on ULTRAFEEDBACK as the dataset itself is also annotated by GPT-4. Some human evaluation can better confirm the superiority of models using ULTRAFEEDBACK.
- Lack of analysis on critique: Given the diversity and complexity of the tasks in ULTRAFEEDBACK, the critique could be task-dependent and complex. More analysis should be presented to help readers to better understand the properties of the critique. The same applies to UltraCM - we only know it generates good critiques but are not sure how useful these critiques are.

**Questions:**

N/A

---

> ### Author Response · Authors · 2023-11-17
> **Response to Reviewer AWP7**
>
> Thank you for the valuable feedback! Here are our responses:
>
> > Q1: Lack of human evaluation
>
> A1: Thanks for the suggestion! although some works showed high agreement between human and model annotations [1,2,3], we must admit that human evaluations are vital. We have conducted a human evaluation on UltraLM-13B-PPO. Specifically, we hired 5 independent annotators to compare model responses. We randomly sample 100 comparison pairs from each benchmark (300 in total), and ask annotators to rank the responses (win/tie/lose). We present human and GPT-4 evaluation results on the 300 pairs in the following table. Overall, human evaluations are mostly consistent with GPT-4 annotations. Against `text-davinci-003`, human annotators gave our model an even higher win rate. Against `gpt-3.5-turbo`, human annotators tended to assign more ties and fewer wins, and the lose rate of our model slightly increased.
>
> | Baseline Model | text-davinci-003          | gpt-3.5-turbo                             |
> | -------------- | ------------------------- | ----------------------------------------- |
> | **Evaluator**  | **Win (%) on AlpacaEval** | **Win/Tie/Los (%) on Evol and UltraChat** |
> | **GPT-4**      | 86.0                      | 61.0/13.0/26.0                            |
> | **Human**      | 89.0                      | 48.7/21.7/29.6                            |
>
>
> > Q2: Lack of analysis on critique
>
> A2: Thanks for the valuable advice. To comprehensively evaluate and analyze the critiques our model generates, we conducted case study in Appendix E.4. On math problem benchmarks, we found that UltraCM-13B could give meaningful critiques on incorrect answers, pointing out flaws and providing suggestions. However, the suggestions are sometimes unnecessarily long, which indicates the imperfection of the model. We will continue to improve the critique quality in future work.
>
>
>
> References
>
> [1] Dubois et al. Alpacafarm: A simulation framework for methods that learn from human feedback. 2023
>
> [2] Lee et al.  Rlaif: Scaling reinforcement learning from human feedback with ai feedback. 2023
>
> [3] Bat et al. Constitutional ai: Harmlessness from ai feedback. 2022

---

### Official Review · Reviewer_GSnG · 2023-11-04

**Soundness:** 4 excellent
**Presentation:** 4 excellent
**Contribution:** 4 excellent
**Rating:** 6
**Confidence:** 4

**Summary:**

This work proposes a large-scale preference data for tuning a LLM with AI preference. Such a dataset can be used to train a reward model, and then used for rejection sampling or RLHF. This dataset can be also directly used for DPO, etc.

**Strengths:**

1. The proposed dataset is large-scale and fine-grained in terms of preference. It is the largest among all current existing preference datasets and the responses come from a wide variety of LLMs.
2. The provided RM training results, best-of-N sampling results, and PPO model training results all show good quality of this dataset.

**Weaknesses:**

1. The contribution of this paper is mainly the proposed dataset, which is collected by widely using the GPT4 API. The method part is all standard method. The collected dataset can be use for research purposes but commercial use is illegal. If ICLR sees a pure dataset contribution paper is appropriate for this conference, then I am ok with accepting it. I am going to give a "weak reject" first and then will check and discuss with AC to see whether I will change my score.
2. Some baseline results are skeptical and inconsistent from what we see in the other papers or leader-board, which I will ask in the "Questions" section in details. These inconsistencies would compromise the integrity of this paper.
3. From Table 2, by comparing the "UltraRM-UF" and "UltraRM-Overall", it looks to me that "UltraRM-Overall" significantly outperforms "UltraRM-UF" in at least 3 out of 4 benchmark test sets, however, this paper claims that fine-grained scores outperform the overall score. For example of this claim "UltraRMOverall discernably lags behind UltraRM-UF and UltraRM on WebGPT. There can be two potential explanations for this observation. First, fine-grained annotation, which scores model outputs from different aspects respectively, provides a more precise assessment for each completion than aggregating evaluation into an overall number.". This claim is controversial given the provided results.

**Questions:**

I feel two parts of baseline numbers are not consistent with public reports:
1. In Table 2, for the StreamSHP model and for the SHP dataset, from this link: https://huggingface.co/datasets/stanfordnlp/SHP, the model should achieve 72.8% on the SHP test set, however, the performance in this work reports as "51.6%". Could you explain the huge gap here?
2. In table 3, for the evolve-instruct and ultrachat test sets, the "Vicuna-13B-v1.5" is shown to be much worse than "LLaMA2-13B-Chat", however, from this public leaderboard: https://huggingface.co/spaces/lmsys/chatbot-arena-leaderboard, the former model is shown to be better than the latter one for both the chat-arena human annotation and MT-Bench GPT-4 annotation.

**Details Of Ethics Concerns:**

This work proposes a dataset, which is collected by extensively using GPT-4 API. Special care may need to be taken for the copyright and terms of use.

---

> ### Author Response · Authors · 2023-11-12
> **Response to Reviewer GSnG (1/2)**
>
> Thank you for the valuable comment!
>
> > Q1: If ICLR sees a pure dataset contribution paper is appropriate for this conference, then I am ok with accepting it. I am going to give a "weak reject" first and then will check and discuss with AC to see whether I will change my score.
>
> A1: As shown in the official ICLR 2024 [website](https://iclr.cc/Conferences/2024/CallForPapers), there is a subject area named *datasets and benchmarks*, which is also exactly the primary area of our paper (Please see above). Therefore, we believe dataset papers are ok for ICLR.
>
>
>
> > Q2: In Table 2, for the StreamSHP model and for the SHP dataset, from this link: https://huggingface.co/datasets/stanfordnlp/SHP, the model should achieve 72.8% on the SHP test set, however, the performance in this work reports as "51.6%". Could you explain the huge gap here?
>
> A2: We are sorry for the ambiguity. The SteamSHP model is different from other reward models in the input format, for it accepts two responses simultaneously (see the model [page](https://huggingface.co/stanfordnlp/SteamSHP-flan-t5-xl)) and outputs which one is better (text-to-text format). During the experiment, we found that there is a position bias issue for this approach, where the reward model tends to prefer the first responses. To eliminate this issue, we average the scores from two runs exchanging response orders to get the final scores, leading to inconsistency. We will add a section in appendix for this issue and present the scores for both runs in the revised paper.
>
>
>
> > Q3: In table 3, for the evolve-instruct and ultrachat test sets, the "Vicuna-13B-v1.5" is shown to be much worse than "LLaMA2-13B-Chat", however, from this public leaderboard: https://huggingface.co/spaces/lmsys/chatbot-arena-leaderboard, the former model is shown to be better than the latter one for both the chat-arena human annotation and MT-Bench GPT-4 annotation.
>
> A3: Thanks for pointing this out! We think this discrepancy may be rooted in the distribution difference of test sets. To comprehensively compare Vicuna-13B-v1.5 and LLaMA2-13B-Chat, we summarize their publicly reported performances from  [Chatbot Arena](https://huggingface.co/spaces/lmsys/chatbot-arena-leaderboard) and [OpenChat paper](https://arxiv.org/pdf/2309.11235.pdf) [1]
>
> |                     | Arena Elo    | MT-bench     | MMLU     | AlpacaEval | MT-bench | Vicuna-bench |
> | ------------------- | ------------ | ------------ | -------- | ---------- | -------- | ------------ |
> | **Metric**          | Human rating | GPT-4 rating | Acc      | Win rate   | Win rate | Win rate     |
> | **Vicuna-13B-v1.5** | **1037**     | 6.57         | **55.8** | 78.8       | 37.2     | 47.1         |
> | **LLaMA2-13B-Chat** | 1026         | **6.65**     | 53.6     | **81.1**   | **55.3** | **86.9**     |
>
> As seen, there is no absolute winner between the two models. Detailed views show that LLaMA2-13B-Chat is more favorable on most conversational questions (AlpacaEval, MT-bench, Vicuna-bench) while Vicuna-13B-v1.5 is better at knowledge QA (MMLU) and real-world conversations (Arena Elo). A plausible guess is that LLaMA-2-Chat models are heavily aligned to be safe [2,3], and it has shown refusal behaviors on many normal questions, which may affect real-world user experiences.

---

> ### Author Response · Authors · 2023-11-12
> **Response to Reviewer GSnG (2/2)**
>
> > Q4: From Table 2, by comparing the "UltraRM-UF" and "UltraRM-Overall", it looks to me that "UltraRM-Overall" significantly outperforms "UltraRM-UF" in at least 3 out of 4 benchmark test sets, however, this paper claims that fine-grained scores outperform the overall score. For example of this claim "UltraRMOverall discernably lags behind UltraRM-UF and UltraRM on WebGPT. There can be two potential explanations for this observation. First, fine-grained annotation, which scores model outputs from different aspects respectively, provides a more precise assessment for each completion than aggregating evaluation into an overall number.". This claim is controversial given the provided results.
>
> A4: Sorry for the confusion. Actually, "UltraRM-UF" refers to the model solely trained on UltraFeedback with fine-grained scores, and "UltraRM-Overall" is trained on UltraFeedback mixed with open-source datasets with overall scores (see Section 3.4). We list the three different versions of UltraRM as below. Therefore, comparing "UltraRM-Overall" and "UltraRM" is the right way to assess the effect of fine-grained scores. We will further clarify the difference in the revised version.
>
> |                     | Data                      | Score        |
> | ------------------- | ------------------------- | ------------ |
> | **UltraRM-UF**      | UltraFeedback             | Fine-grained |
> | **UltraRM-Overall** | UltraFeedback+Open-Source | Overall      |
> | **UltraRM**         | UltraFeedback+Open-Source | Fine-grained |
>
> References
>
> [1] Wang et al. OpenChat: Advancing Open-source Language Models with Mixed-Quality Data. 2023
>
> [2] Touvron et al. Llama 2: Open Foundation and Fine-Tuned Chat Models. 2023.
>
> [3] Zheng et al. LMSYS-Chat-1M: A Large-Scale Real-World LLM Conversation Dataset. 2023.

---

### Author Response · Authors · 2023-11-17
**Summary of Revisions**

We thank the reviewers for their valuable suggestions and constructive comments. Following the reviewers' suggestions, we have revised our manuscript and submitted a new version. In the following, We summarize the primary changes. The revised parts are highlighted in blue color for easier review.

- We further demonstrated the training details of UltraRM-Overall in Section 3.4
- We added sample order experiments for SteamSHP in Appendix D.2 as suggested by reviewer GSnG
- We added human evaluation results in Appendix D.3, which was suggested by reviewer AWP7
- We added a case study for the critiques from UltraCM-13B in Appendix E.4 as advised by reviewer AWP7 and YWNs.

---

### Meta-Review · Area_Chair_GaxH · 2023-12-07

**Metareview:**

This is a borderline paper. The paper presents useful resources, even if methodologically the contribution is thin. However, a critical aspect when relying GPT-4 annotation at this scale and depth is how well it correlates with human judgements. This aspect receives some attention (Section 5.1), but it’s insufficient and lacking. The sample size is tiny. There are also no insights on how human judgements are solicited, what is the agreement among them, and how the agreement with the model breaks down along the different dimensions of annotation. It’s also not clear how this agreement is computed. This is unfortunate, because it seems that this is something the authors considered, just maybe not at the desired depth.

There is also no discussion about how the automated annotation may project biases and other negative predispositions of GPT4. If we imagine this data used to train models broadly, this can lead to harm and unexpected outcomes.

It’s also recommended that the authors expand on the human evaluation of the output models, something they started in response to reviewer AWP7.

**Justification For Why Not Higher Score:**

See above.

**Justification For Why Not Lower Score:**

See above.

---

### Decision · Program_Chairs · 2024-01-16

Reject